# Effect of Inorganic Zinc on Selected Immune Parameters in Chicken Blood and Jejunum after *A. galli* Infection

Viera Karaffová [1,*], Viera Revajová [1], Emília Dvorožňáková [2], Ľubomíra Grešáková [3], Martin Levkut [1], Zuzana Ševčíková [1], Róbert Herich [1] and Mikulas Levkut [1,4]

1 Department of Morphological Disciplines, University of Veterinary Medicine and Pharmacy, Komenského, 73041 81 Košice, Slovakia; viera.revajova@uvlf.sk (V.R.); martin.levkut@uvlf.sk (M.L.); zuzana.sevcikova@uvlf.sk (Z.Š); robert.herich@uvlf.sk (R.H.); mikulas.levkut@uvlf.sk (M.L.)
2 Institute of Parasitology, Slovak Academy of Sciences, Hlinkova, 3040 01 Košice, Slovakia; dvoroz@saske.sk
3 Institute of Animal Physiology Centre of Biosciences of the Slovak, Academy of Sciences, Šoltésovej, 4040 01 Košice, Slovakia; gresakl@saske.sk
4 Institute of Neuroimmunology, Dúbravská cesta, 9845 10 Bratislava, Slovakia
* Correspondence: viera.karaffova@uvlf.sk; Tel.: +421-905-871-840

**Abstract:** Ascaridiosis in poultry results in a reduction in body weight gain, egg production, as well as microelement levels. Infected poultry have higher demands on feed with the addition of essential elements including zinc. The effects of the infection by *Ascaridia galli* and the supplementation of inorganic zinc on the immune status of broilers were monitored through evaluation of the relative expression of selected genes (interleukins, IFN-γ, and TNF-α) by real-time PCR, haematology parameters by microscopy, and quantitative changes of lamina propria lymphocytes by flow cytometry in day 7 and day 14 of the study. We observed that the enrichment of the diet with inorganic zinc has a positive effect on the relative percentage of CD4+ lamina propria lymphocytes in the jejunum and on heterophil counts in blood. In addition, it was concluded that inorganic zinc has an anti-inflammatory effect (downregulation of TNF-α and IL-17) and activates IgA-producing cells in the jejunum of chicks infected with *A. galli*.

**Keywords:** zinc; imunity; *Ascaridia galli*; chicken





## 1. Introduction

In the poultry industry, most chickens with outdoor access are often exposed to a wide range of parasites, e.g., *A. galli*. In this regard, *A. galli* poses a serious biological threat due to its direct life cycle and ability to survive extreme environmental conditions. Ascaridiosis in poultry results in a reduction in body weight gain, egg production, ruffled feathers, drooped wings, high mortality, and other secondary pathological symptoms [1].

Chickens become infected by the ingestion of infective eggs. In the gut's lumen, ingested eggs release larvae, where they molt and stay for approximately 10 days. Larvae gradually penetrate the inner lining of the gut, where they spend 1–7 weeks and molt again. Then, they return to the lumen of the intestine, where they develop into adult worms and the females begin to produce eggs. Larvae in the gut's lining destroy the tissues around them and cause enteritis, which is frequently associated with haemorrhagic exudate. Similarly, adult worms cause mechanical damage of the intestinal wall, thereby contributing to malnutrition. In addition, they compete for nutrients and cause bowel obstruction [2].

Nematodes, such as *A. galli*, activate both cellular and humoral immune responses in the host organism [3,4]. In general, *A. galli* infections have been observed to stimulate classical Th2 immune responses in laying hens [4,5]. Specifically, an increase in serum antibodies IgY and the influx of CD4+ and CD8+ T cells at the site of infection have been recorded during *A. galli* infections [6]. Upregulation of IL-4 and IL-13, but not

IFN-γ gene expression, was observed in chickens' intestine at 14 days post-infection [7]. This contributes to the hypothesis that Th2 polarization predominates during *A. galli* nematode infections. During the migration of parasites in the intestine, pro-inflammatory Th1 response is usually suppressed, which allows, among other things, wound healing of the host [8]. Lambrecht and Hammad [9] reported a positive association between type 2 immunity and IL-17 during eosinophilic inflammation in a mouse model. A similar correlation was found mainly in mice with parasites migrating through the lungs [10,11]. However, another investigation is needed for a deeper understanding of the mechanism regulating *A. galli*-induced Th1 and Th2 immune responses in broilers.

Studies of the relationship between poultry and *A. galli* revealed an adverse impact of *A. galli* on the mineral balance of the host with a reduction in microelement levels in the liver and muscles [12,13]. For this reason, infected poultry have higher demands on feed, with the requirement for the addition of essential elements including zinc.

Zinc (Zn) is an essential mineral involved in many biochemical processes and associated with a wide range of physiological disorders, including weight loss, growth retardation, and nervous and immune system disorders [14]. In addition, zinc is an essential factor in the gene expression of proteins required for growth and development, maintaining cell wall integrity, free radical sequestration, and protection against lipid peroxidation [15].

Zinc deficiency affects cells involved in both innate and acquired immunity at the level of proliferation and maturation. T cell function and the balance between different subsets of helper T-lymphocytes are notably sensitive to changes in zinc levels in the organism. Acute zinc deficiency results in the impairment of innate immunity through the reduction in chemotaxis and phagocytosis of mononuclear cells and has negative impacts on acquired immunity, which causes thymic atrophy with subsequent T cell lymphopenia. Specifically, chronic zinc deficiency results in a significant increase in the production of pro-inflammatory cytokines (for example IL-17), which may initiate the development of autoimmune diseases [16].

Accordingly, the effect of *A. galli* infection and the supplementation of inorganic zinc on the broilers' immune status were monitored through evaluation of the relative expression of selected genes (interleukins, IFN-γ, and TNF-α), white blood cell counts (peripheral blood), and the quantification of immunocompetent cells, such as jejunal lamina propria lymphocytes, on study days 7 and 14.

## 2. Material and methods

### 2.1. Chickens

The chickens were handled and killed according to state regulations. The specific experiment was approved by the Ethics Committee of the Veterinary Medicine and Pharmacy followed by the Committee for Animal Welfare of Ministry of Agriculture of the Slovak Republic (permit number 836/17-221).

A total of 48 chicken broilers of hybrid COBB500 35-day-old were included in the study that lasted 14-days. The birds were randomly divided into 4 groups: control (C), *A. galli* (Ag), zinc (Z), and a combination (*A. galli* + zinc) (Ag + Z). Chickens were fed a commercial diet BR1 (Table 1) ad libitum. The health status of birds was monitored twice a day by visual inspection. Values of room temperature, feed, and water consumption as well as any clinical signs of adverse conditions were recorded daily.

**Table 1.** Composition of commercial diet BR1.

| Ingredients g/kg | BR1 |
|---|---|
| Wheat | 290 |
| Maize | 300 |
| Soybean meal | 320 |
| Rapeseed oil | 40 |
| Fish meal | 20 |
| Limestone | 12 |
| Dicalcium phosphate | 10 |
| Sodium chloride | 2 |
| DL-methionine | 1 |
| Vitamin-mineral mix | 5 |
| **Composition by analysis (g/kg dry matter)** | |
| dry matter | 899.9 |
| crude protein | 232.7 |
| fat | 64.5 |
| dietary fiber | 22.7 |
| ash | 53 |
| Ca | 90.4 |
| P total | 69.6 |

On the second day of the study, birds of the Ag and Ag+Z groups were perorally infected with 500 embryonated *A. galli* eggs in 0.5 mL of phosphate buffered saline (PBS) per bird (via a plastic Pasteur pipette). Zinc and Ag + Z groups were individually subjected to daily peroral administration of aqueous solution of inorganic $ZnSO_4$ at a concentration of 50 mg/0.5 mL PBS from day 1 to day 12 of the experiment. To simulate the same stress manipulation, an equal volume of saline was applied to the control group with a Pasteur pipette. Blood samples were withdrawn by vein puncture from vena subclavia, the chickens were euthanized by intra-abdominal injection of xylazine (Rometar 2%, SPOFA, Prague, Czech Republic) and ketamine (Narkamon 5%, SPOFA) at doses of 0.7 mL/kg body weight, and samples from jejunum were collected during necropsy. Two samplings were performed on days 7 and 14 of the study.

### 2.2. Infective Material and Inoculation

*A. galli* eggs were isolated from adult female worm uterus, obtained from the gut of naturally infected chickens, according to Permin et al. [17], by a gentle mechanical maceration in 0.5 N NaOH. The eggs were embryonated in 0.1 N NaOH in the dark for 4 weeks at 28 °C. During the incubation, the egg suspensions were oxygenated three times a week by stirring. Subsequently, eggs embryonation was evaluated microscopically on a weekly basis starting from day 14. Ultimately, the embryonated *A. galli* eggs were stored in 0.1 N NaOH at 4–6 °C and regularly oxygenated until the application to the experimental chickens.

### 2.3. Homogenization of Jejunum and Isolation of Total RNA of Interleukins (IL-4, IL-17), IFN-γ, and TNF-α Gene

Jejunal samples (*n* = 6) (20 mg weighted pieces) were immediately placed in RNA Later solution (Qiagen, Manchester, UK) and stored at −70 °C before RNA purification and transcription as described in Karaffová et al. [18].

*2.4. Relative Expression of Interleukins, IFN-γ, and TNF-α Gene in Quantitative Real-Time PCR (qRT-PCR)*

The mRNA levels of interleukins, IFN-γ, and TNF-α genes were determined. Moreover, the mRNA relative expression of reference gene, coding GAPDH (glyceraldehyde-3-phosphate dehydrogenase), was selected based on confirmed expression stability using the geNorm program. The primer sequences, annealing temperatures, and times for each primer used for qRT-PCR are listed in Table 2. All primer sets allowed cDNA amplification efficiencies between 94% and 100%.

**Table 2.** List of primers used for the chicken cytokine mRNA quantification.

| Primer | Sequence 5′–3′ | Annealing/Temperature Time | References |
|---|---|---|---|
| IL-4 Fw | AGCACTGCCACAAGAACCTG | 60 °C /30 s | [19] |
| IL-4 Rev | CCTGCTGCCGTGGGACAT | | |
| IL-17 Fw | TATCAGCAAACGCTCACTGG | 59 °C /30 s | [20] |
| IL-17 Rev | AGTTCACGCACCTGGAATG | | |
| IFN-γ Fw | GCCGCACATCAAACACATATCT | 59 °C /30 s | [21] |
| IFN-γ Rev | TGAGACTGGCTCCTTTTCCTT | | |
| TNF-α Fw | AATTTGCAGGCTGTTTCTGC | 59 °C /30 s | [22] |
| TNF-α Rev | TATGAAGGTGGTGCAGATGG | | |
| GAPDH Fw | CCTGCATCTGCCCATTT | 59 °C /30 s | [23] |
| GAPDH Rev | GGCACGCCATCACTATC | | |

Amplification and detection of target products were performed using the CFX 96 RT system (Bio-Rad, Hercules, CA, USA) and Maxima SYBR Green qPCR Master Mix (Thermo Scientific, Waltham, MA, USA). Subsequent qRT-PCR to detect relative expression of mRNA in selected parameters was performed for 36 cycles under the following conditions: initial denaturation at 95 °C for 2 min, subsequent denaturation at 95 °C for 15 s, and annealing (Table 2) and extension step 2 min at 72 °C. A melting curve from 50 °C to 95 °C with readings at every 0.5 °C was produced for each individual qRT-PCR plate. Analysis was performed after every run to ensure a single amplified product for each reaction. All reactions for real-time PCR were conducted in duplicate. We also confirmed that the efficiency of amplification for each target gene (including GAPDH) was essentially 100% in the exponential phase of the reaction where the quantification cycle (Cq) was calculated. The Cq values of the studied genes were normalised to an average Cq value of the reference gene (ΔCq) and the relative expression of each gene was calculated as $2^{-\Delta Cq}$.

*2.5. White Blood Cell Count (WBC)*

One ml of peripheral blood was taken from vena subclavia into Heparin (20 IU.mL$^{-1}$ PBS). A total number of leukocytes were counted using the Bűrker chamber and Fried-Lukáčová solution (475 μL solution plus 25 μL blood) [24]. A white blood cell differentiation was performed, expressed in relative percentages utilising the light microscopy at 1000× magnification on blood smears after staining with Hemacolor (Merck, Darmstadt, Germany), and a count of 100 cells per slide was used. The absolute number of the different types of white blood cell count (G.l$^{-1}$= $10^9 \times$ l$^{-1}$) was determined as follows:

absolute leukocyte count × relative % of a different type of WBC/100 counted cells.

*2.6. Isolation of Lamina Propria Lymphocytes (LPL)*

Isolation of lymphocytes from jejunal mucosa (*n* = 6) was performed by a modification of the method of Solano-Aquilar et al. [25]. At first, the mucin was removed and, subsequently, intraepithelial lymphocytes were isolated [26]. Then, the intestine (cut into 0.5 mm pieces) was washed with 30 mL RPMI-1640 (Sigma, Darmstadt, Germany) for

15 min at 37 °C to remove the previous medium. The supernatant was discarded and the gut fragments were incubated in RPMI-1640 with collagenase type I (15 mg/60 mL RPMI; Sigma-Aldrich, St. Louis, MO, USA) for 1 h at 37 °C. The solution was slightly shaken every 5 min. Collagenase released LPL into the medium. The supernatant fluid was harvested, filtered, and immediately centrifuged at $600 \times g$ for 10 min and resuspended in PBS (Sigma, Germany). Cells were washed two times in PBS (centrifugation 5 min at $250 \times g$) and sediment was resuspended in 1 mL of PBS. Lymphocytes were counted in the Bűrker chamber by Tűrk solution (1:20 ratio) for correct dilution.

### 2.7. Staining of Lymphocytes by Direct Immunofluorescence

After the isolation of bloody and jejunal lymphocytes, their concentration was adjusted to $10^6/50$ μL for immunophenotyping. Labelled mouse anti-chicken monoclonal antibodies CD4, CD8, IgM, and IgA (SouthernBiotech, Birmingham, AL, USA) at protocol-specified concentrations were added to lymphocytes followed by incubation (15 min) in the dark at room temperature. After being stained, the cells were washed once in PBS (centrifugation 5 min at $110 \times g$), resuspended in 0.2 mLof PBS with 0.1% paraformaldehyde, and stored at 4 °C until measurement by flow cytometer.

### 2.8. Flow Cytometry Analysis of Stained Cells (FC)

FACScan cytometer and Cell Quest Program (Becton Dickinson, Heidelberg, Germany) were used to measure and analyse labelled bloody and jejunal LPL subpopulations. Gates were drawn around lymphocytes and the fluorescence data collected on at least 10,000 lymphocytes were analysed by two-parameter dot-plot histogram. The results are therefore expressed as the relative percentage of the lymphocyte subpopulation, which was positive for the specific monoclonal antibodies.

### 2.9. Statistical Analysis

Statistical analysis of data was performed using one-way ANOVA with Tukey post hoc analysis using the statistical program GraphPad PRISM version 6.00. Differences between the mean values for different treated groups were considered statistically significant at $p < 0.05$, $p < 0.01$, and $p < 0.001$. Values in figures are given as means or median in the case of relative gene expression with standard deviations ($\pm$SD).

## 3. Results

### 3.1. Relative Expression for Target Genes

The relative expression for IL-4 gene was significantly upregulated in groups infected by *A. galli* (Ag, Ag + Z) compared to the zinc group and control ($p < 0.05$) on day 14 of study (Figure 1). On the other hand, the relative expression of IL-17 gene was markedly upregulated on study day 7 mainly in Ag group and Ag + Z group than compared to zinc and control groups ($p < 0.01$; $p < 0.001$), as well as on day 14 (Figure 2). Similarly, the relative expression for IFN-γ gene was upregulated in the Ag group compared to zinc ($p < 0.05$) and control ($p < 0.001$), as well as in the Ag + Z group compared to control ($p < 0.001$), and significantly in the zinc group compare to control ($p < 0.001$); however, this was only observed on study day 7 (Figure 3). In a similar manner, TNF-α gene expression was markedly upregulated in the Ag group compared to the other groups ($p < 0.01$; $p < 0.001$) as well as in the combined group compared to zinc and control group ($p < 0.001$) on study day 7. However, on day 14 TNF-α gene expression was downregulated in the jejunum of infected groups compared to the zinc group and control ($p < 0.05$) (Figure 4).

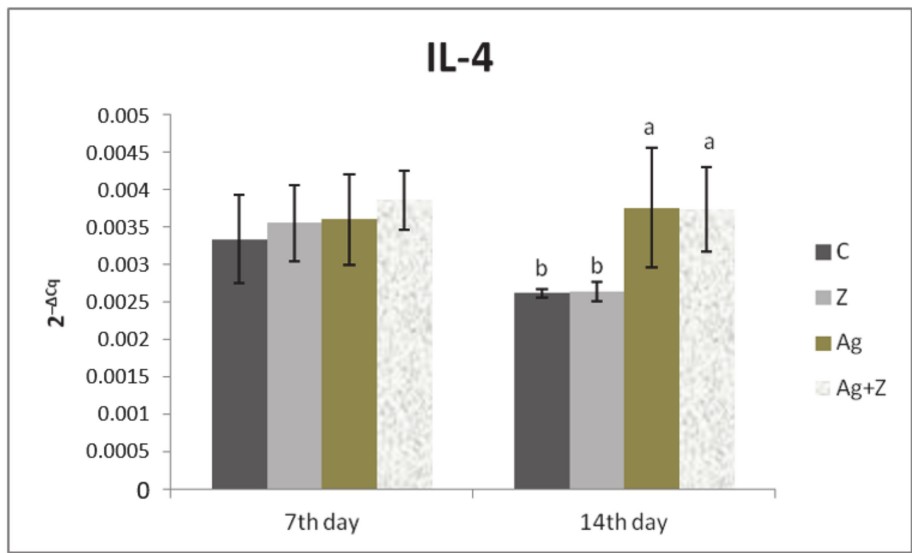

**Figure 1.** Relative expression of IL-4 gene in the jejunum of chickens treated with inorganic $ZnSO_4$ and infected by *A. galli*. Results at each time point are the median of $2^{-\Delta Cq}$. Means with different superscripts are significantly different. [ab] $p < 0.05$.

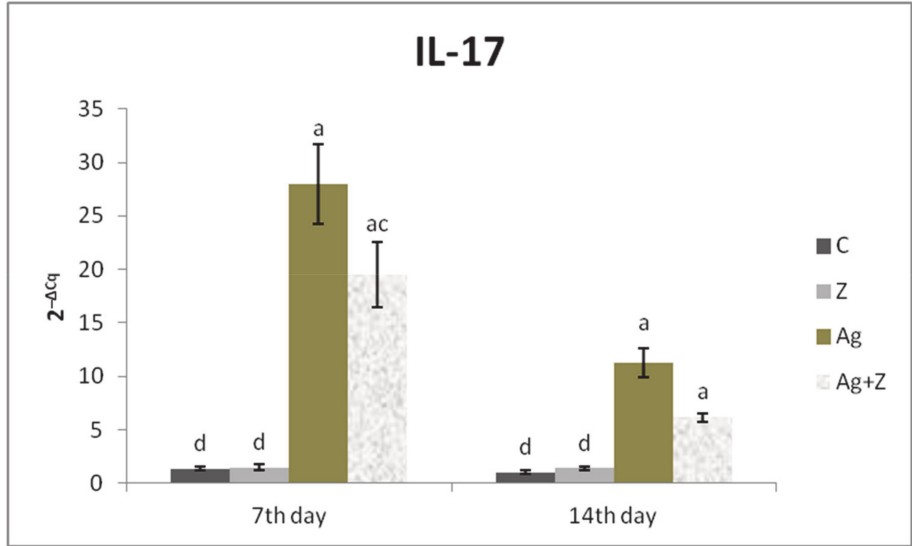

**Figure 2.** Relative expression of IL-17 gene in the jejunum of chickens treated with inorganic $ZnSO_4$ and infected by *A. galli*. Results at each time point are the median of $2^{-\Delta Cq}$. Means with different superscripts are significantly different. [ac] $p < 0.01$; [ad] $p < 0.001$.

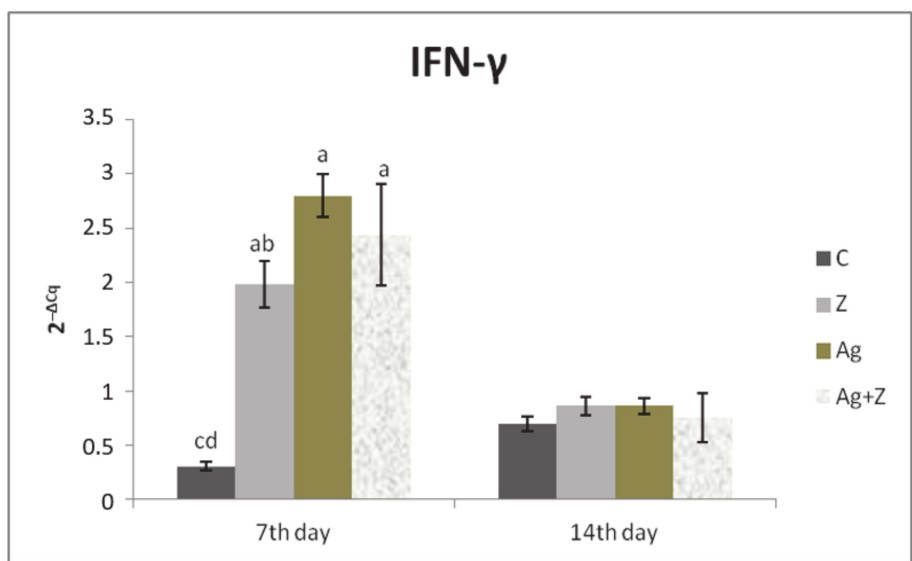

**Figure 3.** Relative expression of IFN-γ gene in the jejunum of chickens treated with inorganic ZnSO$_4$ and infected by *A. galli*. Results at each time point are the median of $2^{-\Delta Cq}$. Means with different superscripts are significantly different. [ab] $p$ <0.05; [ad] $p < 0.001$.

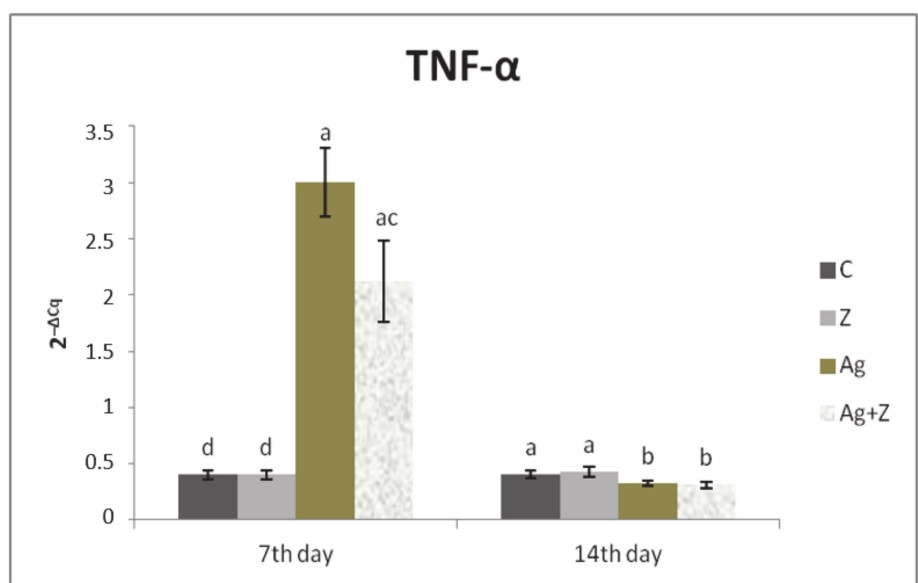

**Figure 4.** Relative expression of TNF-α gene in the jejunum of chickens treated with inorganic ZnSO$_4$ and infected by *A. galli*. Results at each time point are the median of $2^{-\Delta Cq}$. Means with different superscripts are significantly different. [ab] $p < 0.05$; [ac] $p < 0.01$; [ad] $p < 0.001$.

*3.2. White Blood Cell Count (WBC)*

Absolute number of eosinophils in blood of chickens was the highest in the infected group compared to the control group ($p < 0.01$), zinc group ($p < 0.001$), and AG + Z group ($p < 0.05$) on study day 7. Similar significant differences persist on day 14 in comparison with the control group and zinc groups ($p < 0.001$) and the same trend was noticed for the Ag + Z group compared to the zinc group and control ($p < 0.001$) (Figure 5a).

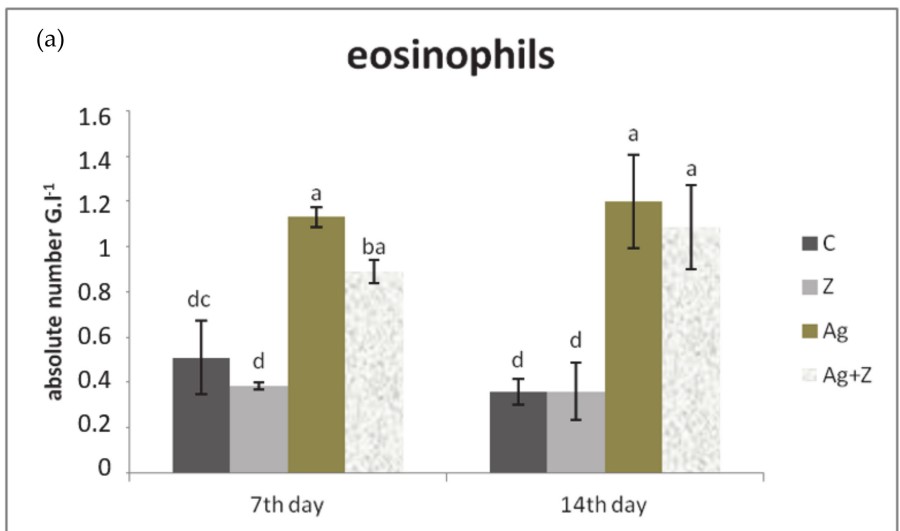

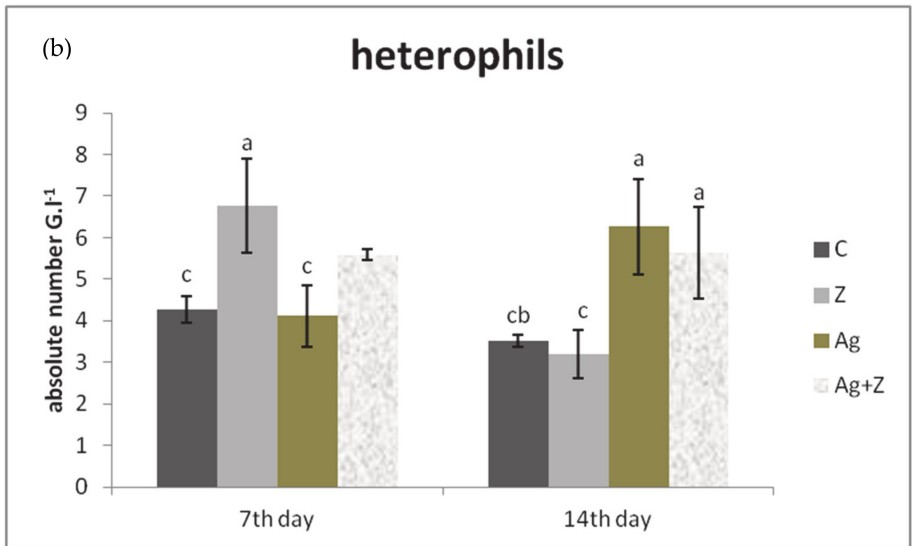

**Figure 5.** Effect of the administration of inorganic $ZnSO_4$ and infection by *A. galli* on the absolute number ($G.l^{-1}$) of (**a**) eosinophils and (**b**) heterophils in peripheral blood. Means with different superscripts are significantly different in Ag group. [ab] $p < 0.05$; [ac] $p < 0.01$; [ad] $p < 0.001$.

On the contrary, the number of heterophils was the highest in the zinc group compared to the control and Ag group ($p < 0.01$) on day 7. In the combined group the number of heterophils was higher compared to the Ag group and control, but not significantly ($p < 0.0085$). During second sampling, the highest number of heterophils was recorded in the Ag group compared to the control and zinc group ($p < 0.01$). Moreover, an increased the number of heterophils was recorded in the Ag + Z group compared to the control ($p < 0.01$) (Figure 5b).

### 3.3. Immunophenotyping of Lymphocytes

The effect of zinc on the relative percentage of CD4+ LPL was manifested in the zinc supplementation group compared to the other groups ($p < 0.001$) and control ($p < 0.01$) on study day 7. On the contrary, on day 14 the relative percentage of CD4+ was significantly improved in the infected group (Ag) compared to the zinc and control group ($p < 0.001$) as well as the combined group ($p < 0.01$) (Figure 6a). The proportion of CD8+ LPL was not significantly influenced by the administration of zinc or *A. galli* infection during both samplings (Figure 6b). The relative percentage of IgM+ cells was markedly stimulated by the combination of zinc and *A. galli* in comparison with the Ag ($p < 0.01$), the Z ($p < 0.001$)

group alone, and the control ($p < 0.01$) on study day 7. On the other hand, the opposite trend for IgM+ cells was noted during second sampling in the Ag group, where it was the highest compared to the control, Zn ($p < 0.001$), and Ag + Zn ($p < 0.01$) groups (Figure 6c). The same tendency was recorded for the relative percentage of IgA+ cells in jejunum in the combination group during the early stages of parasite infection compared to the control and Ag group ($p < 0.001$). On the second sampling, the proportion of IgA+ cells was almost equally highest in the zinc and Ag groups compared to the combination and control groups ($p < 0.01$; $p < 0.001$) (Figure 6d).

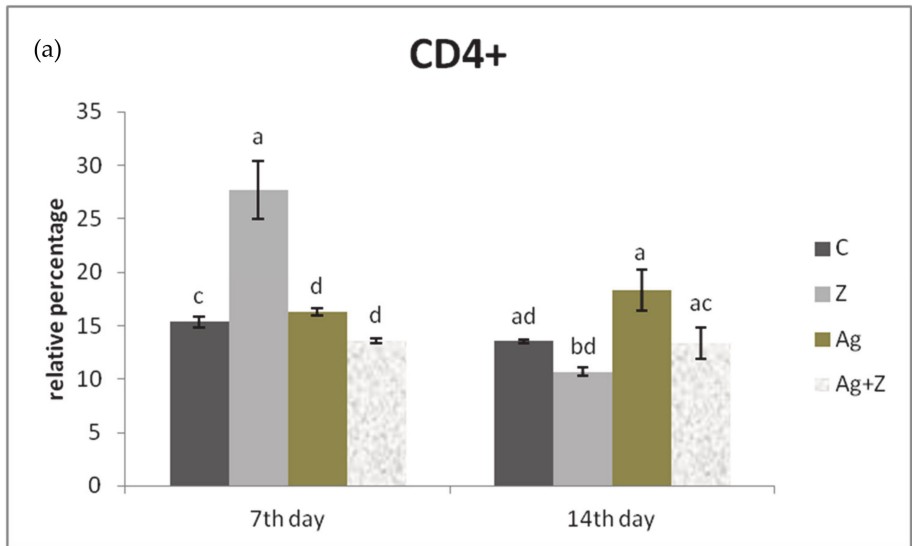

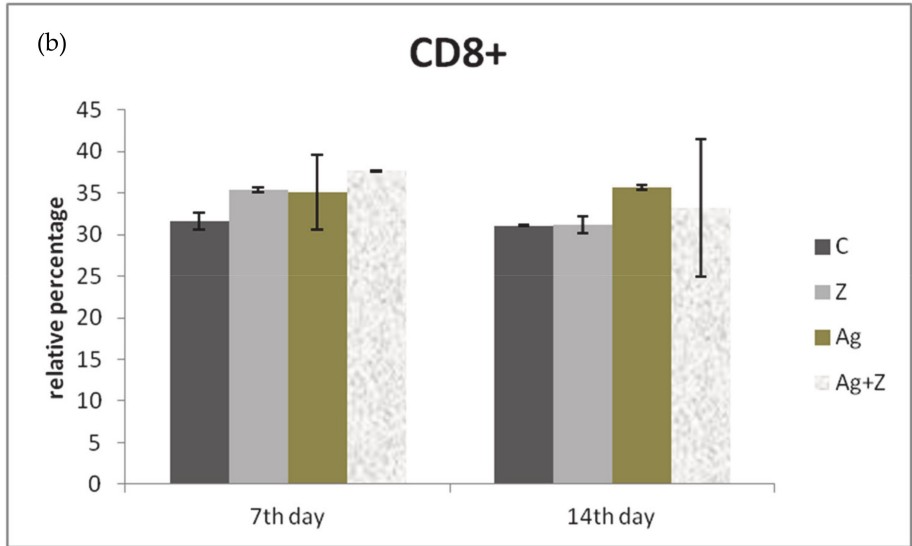

**Figure 6.** *Cont.*

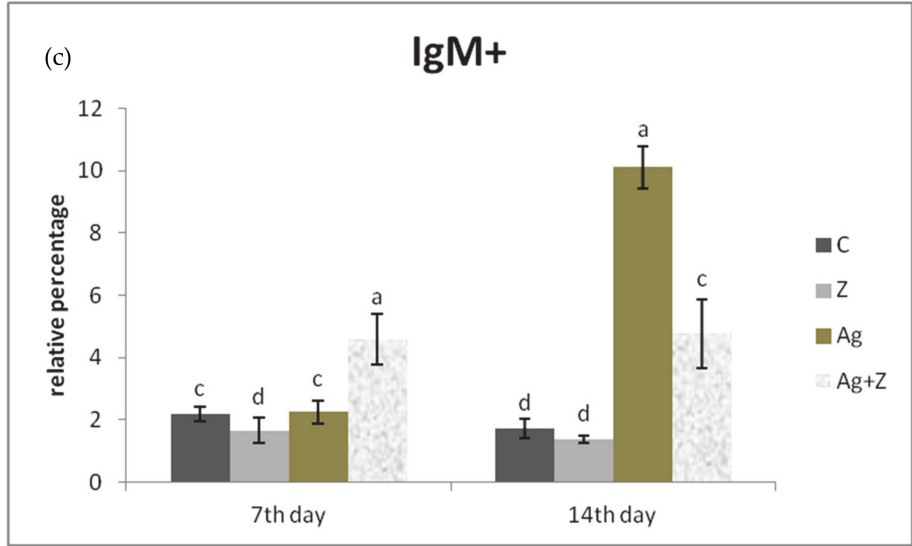

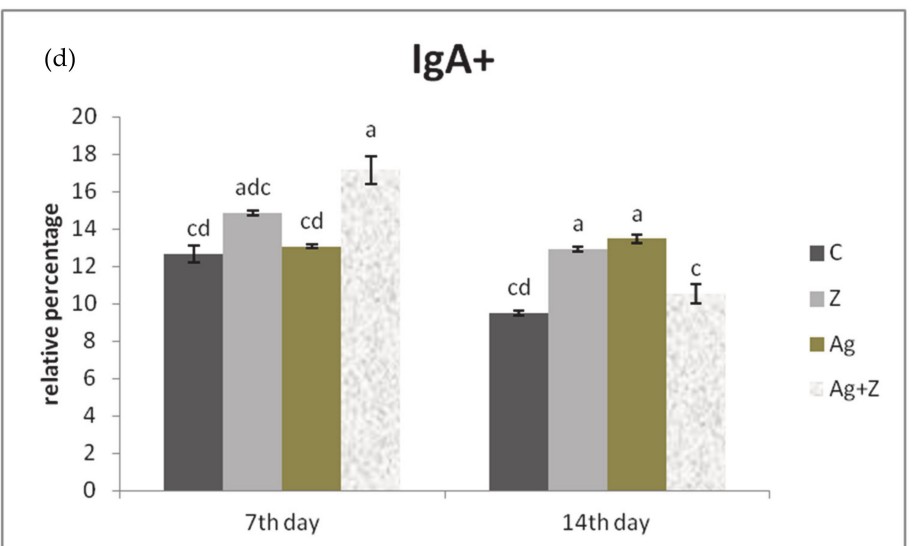

**Figure 6.** Effect of the administration of inorganic ZnSO$_4$ and infection by *A. galli* on the relative percentage of (**a**) CD4+, (**b**) CD8+ (**c**) IgM+, and (**d**) IgA+ LPL in the jejunum. Means with different superscripts are significantly different. $^{ab}$ $p < 0.05$; $^{ac}$ $p < 0.01$; $^{ad}$ $p < 0.001$.

## 4. Discussion

Generally, *A. galli* infections in chickens are accompanied by various clinical manifestations including retarded muscular and osteological development, anorexia, depression, altered hormone levels, and increased mortality [12]. Essentially, the infected chickens are more susceptible to secondary bacterial infections, which may be responsible for the fatal consequences associated with *A. galli* infections [4,27–29]. Therefore, it is necessary to support and modulate the immune response in infected chickens.

In our study, we observed increased IL-4 gene expression in both infected groups on study day 14. This indicates that Th2 cytokines indeed play crucial roles in the intestinal immune reactions during *A. galli* infection in broilers, especially in the later stages of infection [4]. In addition, Finkelman et al. [30] found that IL-4 deficient mice were more sensitive to nematode infections. The anti-inflammatory properties of IL-4 are well-documented along with their ability to suppress classical inflammation, which is necessary for wound repair functions [31].

During the supposed time of larval invasion into the intestinal mucosa (on study day 7), an increased expression of IL-17, IFN-γ, and TNF-α was observed in the infected

groups. This may also indicate the involvement of Th1 component in immune responses during the early phase of infection. On the other hand, during the later stage of infection (day 14), gene expression of pro-inflammatory cytokines (IFN-γ and TNF-α), except for IL-17, was down-regulated in the infected groups and showed the onset of type 2 immunity. Remarkably, IL-17 gene expression was significantly upregulated in the infected groups in both samplings, mainly in the Ag group alone, which suggests close a relationship between the presence of nematode parasites and the activation of Th17 pathway. Furthermore, it could contribute to the pathology of this infection [32]. In the particular case of zinc, its supplementation suppressed IL-17 expression in both samplings. Similar results were presented by Cardenas et al. [33] when zinc supplementation diminished the production of IL-17 by Th17 cells in neonate mice, thereby contributing to the initiation of repair processes.

At the same time, zinc did not have a marked impact on the gene expression of other pro-inflammatory cytokine in the jejunum that may confirm its anti-inflammatory properties. Therefore, zinc supplementation in various forms may be one of the therapeutic options to restore normal mineral balance and simultaneously has a better effect due to its lower utilization and damage to the host's intestinal villi caused by *A. galli* infection [34]. Likewise, Sun et al. [35] observed that zinc deficiency caused shrinkage and flattening of jejunal villi in rats.

Eosinophilia is a characteristic feature of parasitic infections because the larval stages of nematodes can be killed by eosinophils that possess high phagocytic activity [36]. In fact, the onset of the immune response was also confirmed by elevated blood eosinophil levels in both infected groups and this indicates the presence of the parasite in the host. This result is consistent with the study by Tanwar and Mishra [37] and Kumar et al. [38]; they observed an increase in the number of eosinophils in the blood during intestinal helminthiasis, including *A. galli* in poultry. On the other hand, the number of eosinophils was decreased by zinc supplementation in both stages of infection, which confirmed that zinc inhibits the release of the preformed mediators from the basophils and eosinophils [39].

Mainly heterophils are involved in phagocytosis not only in microbial but also during parasitic infections. The phagocytic effect of heterophils, according to the study by Deka and Borah [40], may correlate with their increased amount in the host organism. In our study, infection by *A. galli* revealed the activation of heterophils in the later phase of infection, which could correlate with the destructive activity of the larval stage of *A. galli* in the mucosa of jejunum and the rupture of blood vessels. Similarly, in another study increased heterophils counts were observed on the third day after the infection with *A. galli* [41]. We assume that it also depends on the dose of infection or the immune status of the chickens. As we expected, the effect of zinc in peripheral blood was manifested by an elevation of heterophils during early stage of infection. In contrast, zinc deficiency impaired the oxidative burst of heterophils. It is likely that due to zinc having a central role in the activity of the enzyme superoxide dismutase during oxidative burst, it protects cells from radical oxygen molecules [42]. However, the stimulatory effect of zinc on the total number of heterophils was only temporary. We suppose that the decreased absolute number of heterophils in the Ag+Zn group during the later infection could be caused by a persistent infection, which is when the zinc reserve is depleted due to impaired intestinal absorption.

The most abundant immune cells localised in the subepithelial lamina propria of intestine are lamina propria lymphocytes, which represent the main executive component of the intestinal mucosal immune response. LPL are mostly helper Th-lymphocytes and plasma cells that produce most of the polymeric IgA [43]. In *A. galli*-infected birds we observed increased infiltrations of lamina propria with CD4+ lymphocytes on day 14 of the study. A recent study by Ruhnke et al. [36] reported an increase in intraepithelial CD4+ but a decrease in the number of CD8+ cytotoxic T cell populations after experimental infections with *A. galli* in broiler chickens. In our study, on day 7 we noted only a moderate increase in CD8+ LPL population in the combined group (without statistical significance); however,

it is not clear whether this was due to the influence of *A. galli* infection or inorganic zinc supplementation. In the later phase of infection (day 14), the highest percentage of CD8+ LPL was observed solely in the Ag group, which demonstrates the host's response to intestinal mucosal damage. This is in agreement with the study by Schwarz et al. [4] who showed that, in broiler chickens, a CD8 + T cell response can be expected about 14 days after *A. galli* infection.

On the flip side, an increase in CD4+ LPL recorded during first sampling in the zinc group alone indicates the potential of zinc to induce an immune response. Our results also confirmed the stimulation of the CD8+, IgM+, and IgA+ LPL in the combined group at the first sampling. It has been shown that zinc controls follicular B cell maintenance in the spleen as well as the regulation of the BCR signaling pathway [44–46]. Therefore, the effect of zinc on the humoral immune response is unequivocal. Collectively, the obtained results also point to the activation and cooperation of cellular and humoral immunity in the processes of control of *A. galli* parasites. In addition, zinc supplementation accelerates mucosal regeneration [35].

The highest relative percentage of IgM+ cells was noted in *A. galli* infected group (second sampling) together with the combined group (first sampling). It is correlated with the fact that, at the beginning of the disease process, the level of IgM usually increases, which is stimulated mainly during the first contact of the host with the parasitic antigen.

## 5. Conclusions

Based on our results, it can be stated that the enrichment of the diet with inorganic zinc has a positive effect on humoral and cell-mediated immunity in chickens infected with *A. galli*. In addition, supplementation of inorganic zinc successfully suppressed IL-17 gene expression and, thus, makes a significant contribution to the regulation of potential autoimmune reaction development. Despite the fact that the role of zinc as an important nutritional supplement is well-known, it is still necessary to investigate certain aspects of its use, especially in its application for therapeutic purposes.

**Author Contributions:** Conceptualization, V.K., V.R., Z.Š., M.L. (Mikulas Levkut), and R.H.; methodology, V.K., V.R., E.D., and Ľ.G.; formal analysis, V.R. and M.L. (Mikulas Levkut); data curation, V.K. and M.L. (Martin Levkut); writing—original draft preparation, V.K. and V.R.; writing—review and editing, V.R. and M.L. (Mikulas Levkut); supervision, Z.Š., R.H., and V.R.; funding acquisition, M.L. (Mikulas Levkut) All authors have read and agreed to the published version of the manuscript.

**Funding:** This work was supported by the Grant Agency for Science of Slovak Republic VEGA 1/0355/19.

**Institutional Review Board Statement:** The study was conducted according to the European directive 2010/63/EU on the protection of animals used for scientific purpose and approved by The Ethical Committee of the University of Veterinary Medicine and Pharmacy in Košice and the State Veterinary and Food Administration of the Slovak Republic approved the experimental protocol number 836/17-221 in May 2018.

**Data Availability Statement:** The data presented in this study are available within the article.

**Conflicts of Interest:** The authors have declared no conflict of interest.

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
