# Peer review of "Effect of Inorganic Zinc on Selected Immune Parameters in Chicken Blood and Jejunum after A. galli Infection"

_agriculture, doi:10.3390/agriculture11060551_

Round 1
Reviewer 1 Report
The authors did an effort to answer all the questions and the manuscript was considerably improved. However, I still have some corrections that should be made, and I also found some inconsistencies and English misspelling:
- Introduction
1.1. Line 72: replace "has negatively impact" by "has a negative impact".
- Material and methods
2.2. Line 103: the authors indicated the replacement of " brachial vena" by "subclavia vena ", but no correction was done in the manuscript. Please, check again.
2.3. Line 116: The clarification of the units (g/kg) used for the chemical analysis (crude protein, fat...) in Table 1 that I requested was if it was in g/kg as fed basis or g/kg dry matter. I assume that is g/kg as fed basis…
- Results
3.1. Line 151: the authors indicated a correction of "concentration" by "absolute counts”, but no correction was done in the manuscript.
3.2. Lines 221 to 223: I agree with the sentence "…an opposite trend for IgM+ cells was noted during second sampling in the Ag group". However, you cannot indicate a highly significant P-value between Ag group and all the other groups, as stated by "where it was the highest compared to other groups (p<0.001)". In Figure 6c, no significant difference occurred between Ag and Ag+Zn, as indicated by the letters "a" and "ad", respectively. To avoid confusion, please change the p-value to indicate a tendency (p< 0.05) or you can change the sentence to "where it was the highest compared to control, Zn (p<0.001) and Ag+Zn (p< 0.05) groups".
3.3. Lines 225 to 227: If IgA values were significantly different between Ag and Ag+Z groups, why do you have the letter "a" and "ad", respectively, in Figure 6 d, indicating that these treatments did not differ?
3.4. Lines 229 to 269: The statistics continue to be confusing. I understand your explanation, but things cannot be done that way. Let me give you one example:
Average value from group 1 with letter “ab” versus average value from group 2 with letter “a”. This shows that the means do not significantly differ, even if they are numerically different with some individual values from the group 1 being lower than those from group 2, as indicated by the letter “b”. However, what you should be analysing are the means and not individual values.
Please, correct the following aspects:
Figure 3: Where is the p-value for "b" versus "d"?
Figure 4: no letter is presented above the graph bar for control group (day 14). Is it an "a"? Please, provide the appropriate letter.
Fig 5b: Where is the letter for Ag+Zn group in the graph?
Figure 5: "*" was added to certain letters… Although I understand that “*” indicates significance, this made the Figure more confusing. When there is a significant difference between means, you should indicate a different letter and not an "*". If "a*" is different than "c" (what does not happen between "a" and "c"?), you should change the letter "a" into another letter instead of "a*". By the way, no "c" is displayed in the graph and only "c*" is presented.
3.5. Lines 247 to 250: The letter above the graph bar for Ag+Zn treatment (day 7) is still missing in Figure 5 b (heterophils). The modification made on the manuscript is different from that on the response file. In the Title of Figure 5, the expression "significant in" should be replaced by "significantly different for" and the P-value "acP <0.01" should be presented before "heterophils" and not "eosinophils".
- Discussion
4.1. Line 344: The comma should be removed after "It has been shown..."
Author Response
Revision note
We would like many thanks for comments to our manuscript that help to increase the quality of it. All corrections in the manuscript are marked with yellow background.
We have done the follow corrections by the reviewers:
Reviewer 1:
The authors did an effort to answer all the questions and the manuscript was considerably improved. However, I still have some corrections that should be made, and I also found some inconsistencies and English misspelling:
- Introduction
1.1. Line 72: replace "has negatively impact" by "has a negative impact".
Line 75 we replaced by the reviewer: negatively to negative
- Material and methods
2.2. Line 103: the authors indicated the replacement of " brachial vena" by "subclavia vena ", but no correction was done in the manuscript. Please, check again.
Line 103 we apologize, and now we changed” brachial” to “subclavia”
2.3. Line 116: The clarification of the units (g/kg) used for the chemical analysis (crude protein, fat...) in Table 1 that I requested was if it was in g/kg as fed basis or g/kg dry matter. I assume that is g/kg as fed basis…
Line 93: We corrected in text the sentence: “ Chickens were fed with BR1 diet (Table 1) ad libitum. to the next: “Chickens were fed by commercial diet BR1 (Table 1) ad libitum”.
Line 116: We also corrected the title of Table 1. from Composition of feed mixture BR1 to the next: Composition of commercial diet BR1
Line 116: we also added into Composition by analysis (g/kg) dry mater
as follows: Composition by analysis (g/kg-1 dry matter)
- Results
3.1. Line 151: the authors indicated a correction of "concentration" by "absolute counts”, but no correction was done in the manuscript.
Line 151 I am sorry for mistake. Now we add the correction into manuscript and sentence The concentrations of the different types of white blood count (G.l-1= 109. l-1) were determined as follows:... was changed into: The absolute number of the different types of white blood cell count (G.l-1= 109. l-1) were determined as follows: absolute leukocyte count x relative % of a different type of WBC/100 counted cells.
3.2. Lines 221 to 223: I agree with the sentence "…an opposite trend for IgM+ cells was noted during second sampling in the Ag group". However, you cannot indicate a highly significant P-value between Ag group and all the other groups, as stated by "where it was the highest compared to other groups (p<0.001)". In Figure 6c, no significant difference occurred between Ag and Ag+Zn, as indicated by the letters "a" and "ad", respectively. To avoid confusion, please change the p-value to indicate a tendency (p< 0.05) or you can change the sentence to "where it was the highest compared to control, Zn (p<0.001) and Ag+Zn (p< 0.05) groups".
Lines 221-223 We corrected the sentence "On the other hand, the opposite trend for IgM+ cells was noted during second sampling in the Ag group, where it was the highest compared to other groups (P <0.001) (Figure 6c)" to be more clear as follows: On the other hand, the opposite trend for IgM+ cells was noted during second sampling in the Ag group, where it was the highest compared to control, Zn (P <0.001) and Ag+Zn (P < 0.01) groups (Figure 6c).
3.3. Lines 225 to 227: If IgA values were significantly different between Ag and Ag+Z groups, why do you have the letter "a" and "ad", respectively, in Figure 6 d, indicating that these treatments did not differ?
Lines 225 to 227: Figure 6d was corrected: marking significance letter for Ag+Z was changed from “ad” to “a”.
3.4. Lines 229 to 269: The statistics continue to be confusing. I understand your explanation, but things cannot be done that way. Let me give you one example:
Average value from group 1 with letter “ab” versus average value from group 2 with letter “a”. This shows that the means do not significantly differ, even if they are numerically different with some individual values from the group 1 being lower than those from group 2, as indicated by the letter “b”. However, what you should be analysing are the means and not individual values.
The means was analysed not individual values of group.
Please, correct the following aspects:
Figure 3: Where is the p-value for "b" versus "d"?
The significance for figure 3 in terms with “b” versus “d” for control and zinc group was reanalysed and corrected to “a” (for zinc group) and “d” for control because of P<0.001ad
Figure 4: no letter is presented above the graph bar for control group (day 14). Is it an "a"? Please, provide the appropriate letter.
Yes, letter for control in the second sampling is “a”, thank for your note, corrected figure 4 was insert + L200 was added: “and control”
Fig 5b: Where is the letter for Ag+Zn group in the graph?
The letter is not present because no significance was recorded as we describe in sentence in
Lines 207-208 In the combined group the number of heterophils was higher compared to Ag group and control, but not significantly.
Figure 5: "*" was added to certain letters… Although I understand that “*” indicates significance, this made the Figure more confusing. When there is a significant difference between means, you should indicate a different letter and not an "*". If "a*" is different than "c" (what does not happen between "a" and "c"?), you should change the letter "a" into another letter instead of "a*". By the way, no "c" is displayed in the graph and only "c*" is presented.
3.5. Lines 247 to 250: The letter above the graph bar for Ag+Zn treatment (day 7) is still missing in Figure 5 b (heterophils). The modification made on the manuscript is different from that on the response file. In the Title of Figure 5, the expression "significant in" should be replaced by "significantly different for" and the P-value "acP <0.01" should be presented before "heterophils" and not "eosinophils".
Discrepancies in the figures 5a and 5b were removed both figures were corrected as well as the relevant text in the part of results.
Discussion
4.1. Line 344: The comma should be removed after "It has been shown..."
Line 344 We removed the comma as follows: “It has been shown that zinc controls”

Reviewer 2 Report
Dear author,
you presented the role of zinc on the chicken immune reaction after the A. galli experimental invasion. Presented experiment could reflect on poultry industry and nutrition, the results show the role of zinc in the immune response to A. galli, but still there is need for additional clarifications, as described in specific comments:
L 10 – academy with capitalised letter?
L 20-21 – Zn in eggs and meat? Sentence needs revision
L 21-25 – sentences needs revision;
L 24 – please uniform writing of realtime PCR through the text
L 24 - haematology parameters microscopy?
L35 – please uniform writing of Ascaridia (A.) galli through the text
L 41 – two times „for“
L 59-61 – Ruhnke et al., 2017?
L 91-96 – how old were the chicken when the experiment stared (35 days)? At the end of the experiment they were 49 days old? Why did you start so late with the experiment? Why did the study lasted only 14 days? Did you check the chicken health before starting the experiment (bacteriological or parasitological examination)? What about vaccination protocol?
L 99-101 – why did you choose this dosage of Zn and why this route of administration? references?
L 116 – did the vitamin-mineral mix in the feed contained Zn?
L 119 – why only 6 samples of jejunum were taken? how did you decide on these six samples?
L 121 – the reference mentioned (Karaffova et al., 2019.) refers to the protocol for homogenization of tissue and isolation of total RNA?
L 146 – Heparin? (word missing – tube?)
L 169-170 – protocol specified by the protocols?
L 188 – Results - please provide more information on the chicken health status, morbidity during the experiment, weight gain, etc.
L 210 – the
Line 296-300 sentence need revision. Also, could you please comment the conclusion published by Jarosz et al. (2017) on „the use of zinc in the form of sulphates has no immunomodulatory effect“?
L 389-391 grammatical errors
L 443-444 – is this article written in English?
L 460-462 the title of the article is not correct
Author Response
Revision note
We would like many thanks for comments to our manuscript that help to increase the quality of it. All corrections in the manuscript are marked with yellow background.
Reviewer 2
Dear author,
you presented the role of zinc on the chicken immune reaction after the A. galli experimental invasion. Presented experiment could reflect on poultry industry and nutrition, the results show the role of zinc in the immune response to A. galli, but still there is need for additional clarifications, as described in specific comments:
L 10 – academy with capitalised letter?
L 10 we corrected “Slovak academy of Sciences” to the “Slovak Academy of Sciences”
L 20-21 – Zn in eggs and meat? Sentence needs revision
L20-21 sentence was rewritten.
L 21-25 – sentences needs revision;
L21-25 sentence was rewritten.
L 24 – please uniform writing of realtime PCR through the text
“real-time PCR” was uniform
L 24 - haematology parameters microscopy?
L 24 during correction we omitted by – we add in the text as follows: “haematology parameters by microscopy”
L35 – please uniform writing of Ascaridia (A.) galli through the text
L35 We add the whole name in the abstract as the first mention the name of parasite into the sentence The effect on the infection of Ascaridia galli and supplementation… and then we uniformed all Ascaridia galli into A. galli.
L 41 – two times „for“
L41: we removed from text and substitute “for approximately for 10 days” by “for approximately 10 days”.
L 59-61 – Ruhnke et al., 2017?
No, this sentence is our own reasoning.
L 91-96 – how old were the chicken when the experiment stared (35 days)? At the end of the experiment they were 49 days old? Why did you start so late with the experiment? Why did the study lasted only 14 days? Did you check the chicken health before starting the experiment (bacteriological or parasitological examination)? What about vaccination protocol?
L 91-96 We used in the experiment 35-day-old chicken, as is written in manuscript:
“A total 48 chicken broilers of hybrid COBB500 35-day-old were included in the study that lasted 14-days.”
In organic poultry production the use of floor and aviary systems with free-range areas resulted in a renewed importance for helminthoses of which A. galli with Heterakis gallinarum are the most widespread species. The regular use of antihelmintic treatment is not compatible with organic regulations and alternative control strategies are required. Because we were included in the study with organic and inorganic zinc, we decided to try inorganic zinc for influence to the parasite burden. For this purpose we prepare the experiment with one-month old chickens. During monitoring the literature before the experiment we found some differences in worm population dynamics in age groups, which could be due to innate resistance to A. galli. Some results suggested the better resistance in one month chickens than either very young or old. We were able to observed both, age resistance and zinc administration to the immunity during the histotropic phase of A. galli, It is explanation for 14 lasting experiment. For this purpose we used the chickens from very carefully controlled battery cage system in mentioned age - 35 days. Bacteriology and parasitology were done before experiment and both showed the negative results. All vaccinations were done in hatchery and farm before startin of the experiment.
L 99-101 – why did you choose this dosage of Zn and why this route of administration? references?
L 99-101 The average dose was chosen and used at the base of our previous data published in 2017:
Martin Levkut, Eva Husáková, Katarina Bobíková, Viera Karaffová, Mikuláš Levkutová, Okasana Ivanišinová, Ľubomira Grešáková, Klaudia Čobanová, Katarína Reiterová & Mikuláš Levkut (2017) Inorganic or organic zinc and MUC-2, IgA, IL-17, TGF-β4 gene expression and sIgA secretion in broiler chickens, Food and Agricultural Immunology, 28:5, 801-811, DOI: 10.1080/09540105.2017.1313202
In this paper we published: „One-day-old chickens were fed an unsupplemented basal diet (BD) or the same BD supplemented with 30 or 70 mg/kg of added Zn from ZnSO4•H2O or Zn chelate of glycine hydrate for 40 days“.
The route of administration was applied to exclude the toxicity of solution to the gastrointestinal tract in sense of pathomorphological changes.
L 116 – did the vitamin-mineral mix in the feed contained Zn?
Of course, the vitamin-mineral mix in the feed contained zinc, it's common.
L 119 – why only 6 samples of jejunum were taken? how did you decide on these six samples?
Only 6 jejunal samples were taken because we only had a limited number of animals available.
L 121 – the reference mentioned (Karaffova et al., 2019.) refers to the protocol for homogenization of tissue and isolation of total RNA?
Of course, the protocol for homogenization of tissue and isolation of total RNA is the same as is mentioned and described in “Karaffová et al., 2019”, we would like to avoid repetition.
L 146 – Heparin? (word missing – tube?)
L 146 We prepared heparin ourself in laboratory (diluted in PBS), we have not commercially heparin tubes.
L 169-170 – protocol specified by the protocols?
L 169-170 We corrected Labelled mouse anti-chicken monoclonal antibodies CD4, CD8, IgM, IgA (SouthernBiotech, USA) at protocol-specified in concentrations by the protocols were added... into: Labelled mouse anti-chicken monoclonal antibodies CD4, CD8, IgM, IgA (SouthernBiotech, USA) at protocol-specified in concentrations were added...
L 188 – Results - please provide more information on the chicken health status, morbidity during the experiment, weight gain, etc.
The information about weight gain and other parameters, which are listed in your note, will be published in the other manuscript, in this manuscript we focused on observation chicken health status by measuring of selected immune parameters.
L 210 – the
L 210 we add in front of control the as follows: “compared to the control”
Line 296-300 sentence need revision. Also, could you please comment the conclusion published by Jarosz et al. (2017) on „the use of zinc in the form of sulphates has no immunomodulatory effect“?
I apologise, but the study of Jarosz et al. (2017) is not used in our manuscript, therefore I don’t have reason for comment this study. On the other hand, I suppose that the design of mentioned study (Jarosz et al., 2017) was different.
L 389-391 grammatical errors
L 389-391 5. Darmawi, U.; Balqis, M.; Hambal, R.; Tiuria, F., Priosoeryanto, B.P. Mucosal mast cells response in the jejunum of ascaridia galli-infected laying hens, Media Peternakan 2013, 113–119.
We corrected as follows:
- Darmawi, D.; Balqis, U.; Hambal, M.; Tiuria, R.; Apt, F.; Priosoeryanto, B.P. Mucosal mast cells response in the jejunum of Ascaridia galli-infected laying hens. Media Peternakan 2013, 113–119.
L 443-444 – is this article written in English?
L 443-444 I think you mean the next article:
- Fried, K.; Jantošovič, J. Blood sampling by cardiac punction. Veterinársky časopis 1961, 10: 383–391.
We add after (In Slovak) as is usual in other journals
- Fried, K.; Jantošovič, J. Blood sampling by cardiac punction. Veterinársky časopis 1961, 10: 383–391. (In Slovak)
L 460-462 the title of the article is not correct
The title of article was corrected from “Ascaridia galli infection influences the development 16 of both humoral and cell-mediated immunity after Newcastle disease vaccination in chickens” to “Ascaridia galli infection influences the development of both humoral and cell-mediated immunity after Newcastle disease vaccination in chickens“

Round 2
Reviewer 1 Report
The present manuscript was significantly improved. Well done!
I only have minor comments and suggestions to the manuscript:
- Material and methods
1.1. Line 92: Please replace "fed by" by "fed a"
1.2. Line 115: Please replace "g/kg-1" by "g/kg" in Table 1.
- Results
2.1. Line 209: The statement "...but not significantly." should be followed by a P-value.
2.2. Line 249: Figure 5: Please replace "are significant in Ag group" by "are significantly different".
2.3. Line 253: Figure 5b: Since no significant differences were found between Ag+Zn and the other treatments, the authors should indicate the letter "ac" above the graph bar.
Author Response
All corrections in the manuscript are marked with green background.
The affiliation of co-author Ľubomíra Grešáková was corrected.
Reviewer 1
- Material and methods
1.1. Line 92: "fed by" was replaced by "fed a"
1.2. Line 115: "g/kg-1" was replaced by "g/kg" in Table 1.
- Results
2.1. Line 209: The P- value was added before „but not significantly."
2.2. Line 249: Figure 5 "are significant in Ag group" was replaced by "are significantly different".
2.3. Line 253: Figure 5b: Since no significant differences were found between Ag+Zn and the other treatments, the authors should indicate the letter "ac" above the graph bar
2.3. Line 253: Figure 5b: We disagree, because the superscripts „ac“ mean that the significance between certain groups was P <0.01, but in this case it is not true.

Reviewer 2 Report
Dear author,
thank you for taking into consideration my comments. However, there is still need for minor changes, as follows:
L 25 – sentence still needs revision (in Day 7 and Day 14 of the study)?
L58-60 – there was misunderstanding - „Another investigation is needed for deeper understanding of the mechanism regulating A. galli-induced Th1 and Th2 immune responses in broilers.“. My point was could you maybe related this reference - Ruhnke et al. (2017), to your statement that there is a need for deeper understanding and another investigation?
L388-390 – not corrected in the present version of the manuscript
Author Response
All corrections in the manuscript are marked with green background.
The affiliation of co-author Ľubomíra Grešáková was corrected.
Reviewer 2
L 25 – sentence still needs revision (in Day 7 and Day 14 of the study)?
L 25 –The sentence was corrected: „day“ was added before „7“, to specify on which day sampling was performed
L58-60 – there was misunderstanding - „Another investigation is needed for deeper understanding of the mechanism regulating A. galli-induced Th1 and Th2 immune responses in broilers.“. My point was could you maybe related this reference - Ruhnke et al. (2017), to your statement that there is a need for deeper understanding and another investigation?
Yes, of course it could be related to the reference Ruhnke et al. (2017), because I think, this statement is the general opinion and experience of many researchers in this field of study.
L388-390 – was corrected, specifically „ascaridia galli“ was corrected in italic as „Ascaridia galli“

This manuscript is a resubmission of an earlier submission. The following is a list of the peer review reports and author responses from that submission.
Round 1
Reviewer 1 Report
All the work was performed around an in vivo experiment with clear deficiencies in the design. Among detected deficiencies I want to highlight the low number of animals at each sampling point (3 animals per treatment and sampling point). Furthermore animals in different treatments were submitted to different manipulation programs (peroral infusion in Zn treatment, while control group no manipulated). Human manipulation is a clear stressor for the animals that can be translated to biological responses at immunological level.
No measurements of intestinal damage were performed to verify the extent of the A. Galli administration. No direct evidences about the extent of the infection are shown. I believe this measurements are crucial to verify the experimental infection model worked.
Reviewer 2 Report
The authors have put some effort into the current experiment. However, there are limited experimental units available for the whole investigation. For an animal trial, especially for chickens, the individual variation is pretty high. Hence, the number of samples should be enough.
In the current study, there are some significant issues, especially in statistics.
- How to choose the dose of Zn?
- Individual birds should not be a statistical unit.
- The study design is two factorial designs: (2x2)Zn * challenge. The author should use two–way ANOVA to analyze all the results.
- The sample numbers should be indicated in each result.
- The superscripts are very confusing in terms of mean separation. The mean separation methods did not indicate in the manuscript as well. For example, Figure 2: superscripts: where is b? Figure 3: where is c?, etc., besides, how could one analysis have a different P-value? Is it a contrast test or TT test between each treatment?
I recommend the author re-analyze the data using a more proper way.
Reviewer 3 Report
Dear authors,
The present manuscript presents a significant scientific content with importance for future research. However, I was expecting a higher accuracy in using the English language, as well as more details about the importance of zinc in the immune system of broiler chickens. The authors should also be more careful when writing the Results, in order to avoid inconsistencies.
The authors can find my comments and suggestions below:
Abstract
Line 13: please, specify "microelement levels" in... (probably eggs and meat)
Line 16: please, add by microscopy after haematology parameters
Line 17: replace "(flow cytometry)" by "by flow cytometry"
Line 21: replace "activate" by "activates"
Introduction
Line 36: replace "to cause development" by "and cause"
Line 48: replace "allows among other things, facilitate wound healing of host by "allows, among other things, wound healing of host".
Line 49: replace "published study, where discovered" by "reported"
Line 55: please specify where the reduction in microelement levels occurs (in egg or meat?)
Line 56: replace "involving" by involved
Lines 60-61: "zinc is an essential factor in gene expression". Why is that? Please provide some information about the positive impact of zinc on regulating the expression of selected genes.
Line 63: replace "In the particular case of zinc, its deficiency" by "zinc deficiency".
Lines 64 to 67: "T-cell function and the balance between different subsets of helper T-lymphocytes are notably sensitive to changes in zinc concentration of organism. While acute zinc deficiency causes a decrease in innate and acquired immunity, chronic deficiency improves inflammation". In my opinion, these sentences need to be further detailed. Please, briefly specify the effect of acute zinc deficiency on the decrease in innate and acquired immunity. Does zinc deficiency reduce the production of helper T-lymphocytes? What about the innate immunity?
Line 66: replace "of organism" by "in organism"
Line 68: replace "In accordance with the previous statement" by "Accordingly" and "effect on" by "effect of"
Line 71 to 72: replace "presented by jejuna lamina propria lymphocytes on day" by ", such as jejunum lamina propria lymphocytes, on days"
Material and methods
Line 90: replace "vena" by "vein" or "vena comitans". If you choose to use the Latin word, it should be in italics.
Lines 93 to 94: replace "done first on day 7 and second on day 14" by "done on days 7 and 14"
Line 104: Table 1. Please specify the meaning for the initials (BR1). Is it basal ratio 1? I think it would be better just Basal ratio or Commercial diet. "Composition by analysis (g/kg)" should be "Composition by chemical analysis". The values are expressed as g/kg feed? Please, provide information.
Lines 105 to 130: These subsections of material and methods should be better organised. For instance, you could provide a clearer distinction between "Homogenization..." and "Relative expression..." by preceding them with captions 2.3 and 2.4 with the following text in italics.
Line 107: replace "was" by "were"
Line 108: The authors did not mention that transcription of RNA into cDNA was done, as described in the manuscript [18] by the same authors. Please add "RNA purification and transcription"
Line 116: replace "DNA" by "cDNA".
Line 138: Specify the unit “G.l-1” (G.l-1= 109 l-1?). The results are expressed in concentration or absolute/relative number of cells? Please check formula 1.
Line 168: Add a suitable citation and reference for formula 2. According to the formula, the units are relative number of cells and not relative percentage. Please, check it.
Line 170: Why did you choose Bonferoni post hoc test and not Tukey´s test? Please provide information about the statistical program used (e.g.: SAS or SPSS)?
Results
Line 177: Please, uniform the name of the treatments: Ag+Z instead of AG+Z.
Lines 180 to 184: IFN-γ and TNF-α expressions are up-regulated in both Ag and Ag+Z on day 7. Please, modify these sentences accordingly.
Line 189: replace "the same tendency" by "a similar significant result”.
Line 191: Were the results found with zinc group similar to the ones found with Ag+Z? Provide information in the text.
Line 192: Ag and Ag+Z groups and not only Ag group.
Lines 197 to 199: The relative percentage of CD4+ is higher in Ag than in Zn group but no differences were found between Ag and Ag+Z or control, according to Figure 6 a. Please, check these results and modify the text as suitable.
Lines 203 to 205: I would not consider an "opposite trend". In fact, the percentage of IgM+ cells during day 14 was higher in the Ag group, as the authors described, but only when compared to zinc and control groups. No differences were found between Ag and Ag+Z groups. Please, check these results attending to Figure 6 c and modify the text accordingly.
Lines 205 to 209: These results are confusing. Please, carefully rewrite the sentences attending to the results present in Figure 6 d. For instance, on day 7, the percentage of Ig A in the combination group did not differ from all the other groups, but only between Ag and control. On day 14, zinc and Ag groups led to higher IgA values than control (and did not differ from the combination group).
Line 214: Figure 2 presents the p-values between letters a and ac (correct it as a_acP, as it is written ac), a and d, but not ac and d. Please, add the corresponding p-value to the figure.
Line 218: Figure 3: the p-value between b and d is missing. Please, add it to the figure.
Line 222: Figure 4: The same as in Figure 2 for the P-values. Also, the letter for control is missing (probably a?).
Lines 210 and 225: Fig. 1, 2, 3 and 4: The results are the median of 2–ΔCq or the average of 2–ΔCq? Please, modify it accordingly. The units in the Y axis should be presented as “2–ΔCq” instead of “2 delta Cq”.
Lines 226 to 232: Figure 5 a and b: The P-values (abP and acP) are provided but no distinction was made between a_abP and a_bcP and a_cP and a_cdP. Please, provide that information. Figure 5 b: the letter for significant differences is missing for Ag+Z treatment on day 7. Lines 230 to 231: absolute/relative number or concentration? Please, define better the units used in Figure 5 legend, as well as in the Y axis of the graphs. See comments on line 138.
Lines 233 to 243: Please, see previous comments on the other Figures about the missing P-values. The units are relative percentage or relative number? See comments on line 168.
Discussion
Line 247: replace “infested” by “infected”.
Lines 271 and 272: "At the same time, zinc did not have marked impact on the gene expression of other pro-inflammatory cytokine in jejunum that may confirm its anti-inflammatory properties." I do not agree with the sentence as the present results clearly show a significant impact of zinc on the gene expression of other pro-inflammatory cytokine, such as a suppression of TNF and IFN at the first stage of infection and IL-4 at a latter stage.
Lines 274 to 275: "also has a better effect because of it lower utilization by A. galli infected host organism [35]. This "better effect" deserves to be further explored. I understand that the infected chicken would present a disruption of their intestinal villi and, thus, the zinc would be less absorbed and used by the host. This sentence needs to be rewritten.
Line 274: replace it by its.
Line 283: replace through by in.
Line 288: "In further study, the authors". It seems that the study was a posterior study and that the authors are the same of the present manuscript. Please delete this and rewrite the sentence.
Lines 293: "zinc deficiency impaired the oxidative burst of heterophils". Why did this happen? This aspect deserves to be better explored in the text.
Line 292: Please delete "peripheral blood". It is a repetition.
Lines 294 to 295: "However, stimulatory effect of zinc on the total number of heterophils was only temporary". This sentence should appear after the sentence between lines 291 to 291. "...early stage of infection. However, stimulatory effect..." Why this effect did not occur in a latter stage of infection? In my opinion, you should also discuss the effect of zinc on eosinophils, which were decreased by zinc supplementation in both stages of infection.
Line 308: Please delete "good".
Line 316: Replace "leads to accelerated...and increased..." by "accelerates...and increases"
Line 317: "intestinal enzymes...". What are these enzymes? This part appears without context.
Lines 311 to 321: Why do the levels of IgM and IgA increase with the supplementation of infected chickens with zinc in the first stage of infection? While I understand that zinc stimulates the immune system of broilers and, thus, boosts an immune response, the effect of zinc on humoral immune response needs to be further explored. Please, cite some studies where the zinc influences the levels of IGs. Can the zinc act on the expression levels of IGs with a consequent increase of their production?
References
Please delete “1. References” in line 339 and all the other numbers that appear before each reference.